# Thymus in Cardiometabolic Impairments and Atherosclerosis: Not a Silent Player?

**DOI:** 10.3390/biomedicines12071408

**Published:** 2024-06-25

**Authors:** Irina V. Kologrivova, Natalia V. Naryzhnaya, Tatiana E. Suslova

**Affiliations:** Cardiology Research Institute, Tomsk National Research Medical Center, Russian Academy of Sciences, 111A Kievskaya, Tomsk 634012, Russia; natalynar@yandex.ru (N.V.N.); tes@cardio-tomsk.ru (T.E.S.)

**Keywords:** thymus atrophy, thymus adipose tissue, atherosclerotic plaque, atherogenesis, T lymphocytes, inflammaging

## Abstract

The thymus represents a primary organ of the immune system, harboring the generation and maturation of T lymphocytes. Starting from childhood, the thymus undergoes involution, being replaced with adipose tissue, and by an advanced age nearly all the thymus parenchyma is represented by adipocytes. This decline of thymic function is associated with compromised maturation and selection of T lymphocytes, which may directly impact the development of inflammation and induce various autoinflammatory disorders, including atherosclerosis. For a long time, thymus health in adults has been ignored. The process of adipogenesis in thymus and impact of thymic fat on cardiometabolism remains a mysterious process, with many issues being still unresolved. Meanwhile, thymus functional activity has a potential to be regulated, since islets of thymopoeisis remain in adults even at an advanced age. The present review describes the intricate process of thymic adipose involution, focusing on the issues of the thymus’ role in the development of atherosclerosis and metabolic health, tightly interconnected with the state of vessels. We also review the recent information on the key molecular pathways and biologically active substances that may be targeted to manipulate both thymic function and atherosclerosis.

## 1. Introduction

The thymus, a primary organ of the immune system, represents a “nursery” for all the T lymphocytes in the human body. It has mesodermal origin and fully differentiates by the 17th week of embryonic development [1]. Early thymic progenitors (ETP) migrate to the thymus from bone marrow, where they undergo differentiation into mature naïve CD4+ or CD8+ T lymphocytes [2]. The thymus is actively functioning in infancy but undergoes involution in adulthood, being nearly totally replaced with adipose tissue. Meanwhile, thymic functionally active regions may be found in humans even at an advanced age [3]. Thymic functional activity in adulthood is interconnected with the degree of immune senescence, susceptibility to infections, and development of various non-infectious diseases, including atherosclerosis [4,5].

According to epidemiological data, the spread of atherosclerosis increases with age and corresponds to the degree of thymus degradation [5]. The lipid-based concept of atherosclerosis pathogenesis has been left behind, and currently atherosclerosis is unequivocally regarded as an immune-based disease [6,7,8]. At the same time, current therapeutic approaches to control atherosclerosis progression and prevention of its complications are still centered on the control of lipid metabolism. Only a few clinical trials directed towards the control of inflammation have demonstrated serious inspiring results so far [9,10]. The success in control of unfavorable cardiovascular events in each of these studies was associated with increased frequency of microbial infections, as they were all associated with suppression of inflammation.

The perspective not only to suppress inflammation during atherosclerosis but rather to regulate its development through normalization of the intrinsic mechanisms is very attractive and desirable. The possibility to prolong the functionality of thymus and normalize its regulatory role in maturation of immune cells seems a very tempting therapeutic approach. Unfortunately, data on the thymic function in adults and its involvement in atherogenesis remain scarce and the role of the senescent thymus in pathogenesis of cardiometabolic disease is majorly underappreciated.

In the present review, we have undertaken an effort to summarize existing data on the mechanisms involved in thymus involution, role of thymus in atherogenesis, signaling pathways shared between thymus functionality and atherosclerosis, existing methods of thymic function evaluation, as well as therapeutic approaches targeting thymus that could have found a place in cardiology in the nearest or distant future.

## 2. Basics of T Cell Selection in Thymus

The term thymus stems from the Greek word “thumos”, that can be translated as “principle of life; heart; soul; passion”. However, the role of the thymus in immune response was not recognized until 1960s [4].

The thymus represents a primary organ of immune system, responsible for the generation and maturation of T lymphocytes (helper T cells, cytotoxic T cells, and regulatory T lymphocytes). The key events that take place in the thymus are positive and negative selection of thymocytes. During maturation, naïve T cells are checked for their ability to bind major histocompatibility complex (MHC) or human leukocytes antigen (HLA) in humans. Cells that demonstrate this function successfully are positively selected, or, in other words, receive a signal for further survival [5]. Medullary thymic epithelial cells (mTECs) of young thymus have a unique ability to express genes of the tissue-restricted antigens (TRA), which promote exposure of the maturing T cells to the autoantigens that they may encounter further-on in the host’s organism. In case of the strong interaction of the definite T cell receptor (TCR) and TRA, the clone expressing this specific TCR undergoes negative selection through apoptosis and does not leave the thymus. Another scenario is skewing of T cell differentiation towards T regulatory lymphocytes, which will suppress immune response upon recognition of this particular antigen. This process is known as central tolerance and protects organisms from the development of the potentially dangerous autoimmune reactions [11,12]. The ability of mTECs to express up to 90% of the coding genome provides protection to the host organism against attack by its own immune cells [13]. Expression of TRA is controlled by the gene *Aire*AutoImmune REgulator), expressed by mTECs [14].

Recently, another transcription factor, Fezf2 (forebrain embryonic zinc-finger-family zinc finger 2), has been demonstrated to be responsible for expression of tissue-restricted antigens in the thymus, mainly those expressed in the lung and the liver [15,16]. The spectrum of proteins, regulated by Fezf2, differs from those regulated by AIRE [15].

## 3. Thymus Involution

In humans, thymus senescence starts by the second year of life, and progressively continues thereafter until nearly total atrophy [14]. By the age of 45 years, adipose tissue constitutes 75% of the thymus volume [17], morphologically consisting of multiple lipid-laden multilocular cells (LLMC) with proinflammatory properties [18]. It would be a mistake to consider thymus in elderly individuals as a useless organ. It has been demonstrated that removal of the thymus during surgery with sternotomy was associated with increased rate of total mortality, higher frequency of autoimmune disorders, decreased production of naïve CD4+ and CD8+ lymphocytes, and elevated levels of inflammatory cytokines in blood [19].

Physiological thymus involution is divided into two phases: early phase (growth-dependent involution, starts early in life and evolved most likely as a protective mechanism, required for energy redistribution) and late phase (age-dependent involution, coincides with aging in other organs) [20]. Acute thymic involution may also take place. It is induced by different microbes, glucocorticoids, radiation, or use of chemotherapy and is associated with development of inflammation [21]. It has been demonstrated that premature thymus atrophy is at least partially caused by the damage of stromal cells by oxidative stress, associated with the deficiency of catalase, an important enzyme-antioxidant catalyzing reaction in which hydrogen peroxide turns into water and oxygen [22]. Free cholesterol and ceramides are also potential agents inducing thymic involution.

Even though the replacement of immune cells in the thymus with adipose tissue with age was discovered long ago, the process of thymic adipogenesis remains not fully understood. The oversimplified hypothesis stating that fat just passively occupies the niches abandoned by thymocytes appeared to be non-competent, since animals with lymphopenia did not develop thymic adiposity [13].

There are several theories of physiological thymic adipogenensis. Neither of them can be rejected, just as neither can be accepted without hesitation. First is that cells of the perivascular space, such as mesenchymal stem cells, perivascular cells, and pericytes, differentiate into adipocytes through the activation of peroxisome proliferator-activated receptors (PPARγ), the main adipogenic transcription factor [21].

Another hypothesis is that adipose cells arise from TECs. The process presumably consists of two steps: first, endothelial cells turn into mesenchymal cells (the process received the name epithelial-mesenchymal transition, EMT), and only after that do mesenchymal cells undergo adipogenesis [20]. During age-dependent thymic involution, the spatial distribution of adipocytes was not random but corresponded to the distribution of former mesenchymal cells [23], which also confers their involvement in adipogenesis. Thymic adipocytes and the remaining thymocytes are most likely interacting with each other through the paracrine signaling [24].

Tissue growth factor (TGF)-β plays the key role during EMT, as mice lacking TGF-beta receptor on their TECs demonstrated impairment of thymic physiological senescence [25]. TGF-β decreases the intracellular transcription of E-cadherin, an epithelial marker [26]. Markers of mesenchymal cells such as vimentin, N-cadherin, and fibronectin, on the contrary, become elevated [21,27].

Adipogenic thymus involution appeared to be dependent on the interaction between TECs and T lymphocytes. Depletion of molecule CD147 (also known as EMMPRIN, extracellular matrix metalloproteinase inducer) on T lymphocytes inhibited EMT in mice [26]. Effects of CD147 on TECs required the presence of TGF-β and involved degradation of E-cadherin [26].

Fibroblast growth factor (FGF)-21, highly expressed in TECs, on the contrary, decreased adipogenesis in thymus and supported maintenance of thymic cellularity [17]. External factors, such as s caloric restriction, reduced expression of peroxisome proliferator-activated receptor (PPAR)-γ, the key transcription factor of adipocytes, and also suppressed thymic adipogenic involution [28].

Small non-coding RNA molecules (microRNAs) also appear to be involved in regulation of adipogenesis in the thymus. The expression of miR-181a-5p was decreased in aged thymus. Most likely, it suppressed the expression of TGF-β receptor and thus supported proliferation of TECs [29]. The EMT process was also suppressed by miR-200 and miR-205 [30].

The schematic thymus involution through the replacement with adipose tissue is represented in Figure 1.

The process of replacement of functional thymocytes and TECs with adipose tissue seems to be associated with development of chronic low-grade inflammation (inflammaging). Activation of NLRP3 inflammasome (nucleotide-binding domain, leucine-rich–containing family, pyrin domain–containing-3 inflammasome) and the subsequent release of interleukin (IL)-1β and IL-18 was an indispensable part of thymus functional decline. Thymic myeloid cells (macrophages and dendritic cells) were the main producers of IL-1β while IL-1β receptors were mainly expressed on TECs [31]. Large lipid and protein aggregates in macrophages, possibly formed due to the defective autophagy in cells, could be possible activators of NLRP3-inflammasome [31].

## 4. Thymus and Metabolism

Availability of various metabolic nutrients and metabolic microenvironment impacts the polarization of immune response. This observation has even led to identification of the definite research field, termed “immunometabolism” [32]. Active metabolic compounds have also been demonstrated to influence thymus function.

Close interconnections between thymus and metabolic health exist at several levels. Thus, newborns of women with gestational diabetes developed DiGeorge Syndrome (associated with deletion of 22q11.2 and absence or underdevelopment of the thymus) [33]. Patients with higher body mass index (BMI) developed immune-related adverse events after prescription of immune-checkpoint inhibitors, affecting the process of negative selection in the thymus more readily than patients with lower BMI [34].

Thymic function is highly dependent on the adipokines (cytokines produced by adipose tissue) leptin and ghrelin [4]. Administration of leptin had stimulatory effect on the thymus, while leptin-deficient ob/ob mice were characterized by severe thymus atrophy [35]. Transplantation of adipose tissue from wild-type mice successfully restored immune impairments [13]. Direct effects of other adipokines, such as apelin or adiponectin, on thymus cellularity has not been demonstrated yet, but are very plausible. Apelin receptor was discovered in the thymus [36], while Treg-lymphocytes in thymic nurse complexes expressed adiponectin, which regulated maturation of T cells [37]. The main adipokines involved in regulation of thymic function are summarized in Figure 2.

On the other hand, chest traumas with thymus injury in adults were associated with the development of premature immune senescence and increased risk of obesity, metabolic syndrome, and atherosclerosis [4,38].

In the rat model of diet-induced obesity, it was demonstrated that obese animals developed a larger thymus with increased cellularity, perhaps through various growth factors produced by adipose tissue, including insulin-like growth factor (IGF). However, these obese rats also tended to develop age-associated thymus involution more rapidly than their age-matched counterparts. The proportion of naïve T lymphocytes in the periphery also correlated with thymus local cellularity: obesity was associated with increased T cell memory conversion and lymphocytes exhaustion [39]. Tregs were produced in more abundance than effector T lymphocytes, which significantly compromised the ability to fight tumor growth in aged animals.

According to the data of computed tomography (CT), complete fatty degeneration of thymus was associated with male sex, higher BMI, hypertension, dyslipidemia, sedentary lifestyle, and lower fiber intake, and was accompanied by the impaired ability to produce naïve T cells [1].

Since obesity is usually associated with elevated leptin levels, the most likely thymus impairments may be caused by the development of leptin resistance or inhibitory effects of glucocorticoids, which may be additionally produced in adipose tissue during obesity [13]. Estrogens, which are highly produced by aromatase in white adipose tissue, also inhibited thymopoiesis, promoting apoptosis both in thymocytes [40] and thymic epithelial cells [41], and suppressed expression of AIRE and TRA in thymus [42].

## 5. Thymus and Atherosclerosis

Interconnection between atherogenesis and thymus involution is not that obvious and stems primarily from experimental studies. However, Huang S. et al. (2016) demonstrated that the number of the signal-joint T cell receptor excision circles (sj-TREC) in T lymphocytes, a surrogate marker of the thymic function, was decreased in patients with unstable and stable angina compared to unaffected individuals. Of note, the number of sjTREC was also inversely correlated with the severity of atherosclerosis and was the lowest in patients with high Gensini Score (a cumulative measure of atherosclerosis severity according to coronary angiography results) exceeding 41 points [43].

Expression of tissue restricted antigens in thymus has been shown to decline with age [12], which prompted the idea of the potentially declined level of negative selection. Hester A.K. et al. demonstrated that age-dependent atrophy is accompanied by the increase of autoreactive clones of T lymphocytes in the periphery [11]. Expression of tissue restricted antigen associated with atherosclerosis, namely apolipoprotein A (ApoB), also declined in thymus with age. As a result, ApoB-specific clones of T lymphocytes escaped thymus and represented a potential auto-aggressive cell population [11]. Some T lymphocytes expressed mixed T regulatory and T effector phenotype, being able to produce IFN-γ upon stimulation and exacerbate the development of atherosclerosis [44]. Overexpression of mitochondrial catalase, even though decreased oxidative stress in thymus stromal cells and reduced thymus atrophy, did not influence expression of ApoB in aging thymus, and did not impact the level of the inefficient negative selection [11,22].

It was the balance between ApoB-specific conventional and regulatory T lymphocytes reaching arterial wall that determined susceptibility of apolipoprotein E deficient (ApoE−/−) mice to atherosclerosis [45]. Of note, the production of ApoB-specific clones of T lymphocytes did not depend on transcription factor AIRE in mice [46].

Mice with hypertension, one of the most important risk factors of atherosclerosis, were characterized by the reduction of thymus function [47]. Hypertension in these animals was associated with Th1/Th2 imbalance, which was restored upon transplantation of thymus from neonatal mice [47].

There is also a hypothesis that low-density lipoproteins cholesterol, a potent factor of atherosclerosis, may reduce an important thymic regulatory factor, Foxn1, leading to decline of the thymic function [5].

In our previous studies, we have demonstrated interconnection between the secretory activity of thymic adipose tissue and arterial vascular stiffness. Parameters of vascular stiffness were directly related to the degree of leptin secretion by thymic adipocytes and, indirectly, to insulin secretion [48].

Data on the consequences of thymectomy in adults are very scarce. The only work we came across was a retrospective study by Kooshesh K.A. et al. (2023) [19] who demonstrated that patients who had undergone removal of thymus had an increased rate of autoimmune disorders in the next 5 years of follow-up compared to patients without thymus removal. These results were obtained only after patients with cancer, postoperative infection, or pre-existing autoimmune conditions were excluded from the analysis [19]. Development of atherosclerosis was not considered in this study. However, in our opinion this paucity of data only emphasizes the necessity of the role of thymus in patients with cardiovascular disorders.

The summarized evidence of the protective functions of thymus in atherosclerosis is summarized in Table 1.

## 6. Thymic T Regulatory Lymphocytes Control Atherogenesis

As we have already mentioned, one of the main functions of the thymus is generation of anti-inflammatory immune suppressive T regulatory (Treg) lymphocytes. The decline of Tregs capacity to control the development of immune response is one of the major consequences of thymus involution, affecting progression of atherosclerosis.

The master transcriptional regulator of Tregs is FoxP3 (forkhead box P3) [55]. CD25+FoxP3+ Treg lymphocytes originate from the thymus and migrate to the periphery, where they can recognize self-antigens and suppress the development of immune response through contact interaction with effector T cells and antigen-presenting cells or through the secretion of cytokines: TGF-β, IL-10, IL-35 [56]. Thus, Tregs support tolerance in the periphery, inactivating clones of T cells that, for any reason, could have evaded negative selection in the thymus. It was demonstrated that Tregs may also re-circulate back to thymus via chemokine signaling, where, without additional stimulation with AIRE, they suppress the development of Tregs de novo through the consumption of the key developmental cytokine—IL-2 [57]. Treg lymphocytes may also arise in the periphery from the naïve T cells, but expression of FoxP3 in them is unstable and dependent on the cytokine and growth factors milieu [58]. A transcription factor, Helios, allows us to distinguish thymic-derived Tregs from peripherally induced Tregs [59]. Generation of Tregs in the thymus was dependent upon AIRE [60].

In Table 1 we presented studies describing the involvement of thymic Tregs in pathogenesis of atherosclerosis as one of the protective functions of the thymus during coronary artery disease. The first evidence of the involvement of Tregs in pathogenesis of atherosclerosis was obtained in 2006, when it was demonstrated that ApoE−/− mice depleted with Tregs developed larger and more vulnerable atherosclerotic lesions [50]. In LDLR−/− mice, depletion of Tregs led to increased vascular inflammation and elevation of pro-atherogenic lipids in circulation [51]. Tregs may reduce inflammation in endothelial cells, thus modulating endothelial dysfunction. In HUVEC cells, Tregs suppressed activation of nuclear factor kappa-light-chain-enhancer of activated B cells (NF-κB) signaling pathway, as well as down-regulated expression of vascular cell adhesion protein 1(VCAM-1), monocyte chemotactic protein 1 (MCP-1), and IL-6 [61]. Through contact interaction with effector T lymphocytes via their surface ectonucleotidase CD39, Tregs suppressed adhesion of inflammatory cells to endothelium [62]. Regression of the atherosclerotic plaque was associated with accumulation of Tregs [63].

In a large prospective study, it was demonstrated that decreased numbers of CD4+FoxP3+ Treg cells in the peripheral blood are associated with the development of myocardial infarction in patients [53]. In patients with non-ST elevation acute coronary syndrome, the reduction in the numbers of Tregs occurred primarily due to the impaired thymic export of these cells, as memory Tregs were not impaired. These changes were accompanied by the increased levels of inflammation [52]. The data of the involvement of primarily thymic Treg cells in the development of acute coronary syndrome were also confirmed in the study of Jiang L. et al., who demonstrated that Helios+ Treg cells were reduced in patients with acute coronary syndrome compared to patients with stable angina and healthy control [54].

There are various approaches to normalize Treg function and thus modulate the development of atherosclerosis. However, all of them are currently at the preclinical stage. Treg lymphocytes delayed the development of atherosclerosis upon their activation via intranasal, oral, or subcutaneous introduction of any of the atherosclerosis associated antigens [56]. Oral administration of oxLDL or malondialdehyde-treated LDL to LDL receptor−/− mice fed Western-type diet increased numbers of FoxP3+ Tregs in both the spleen and mesenteric lymph nodes and was associated with decreased initiation and progression of atherosclerosis [64]. Oral administration of heat shock protein (HSP)60 to LDL receptor−/− mice induced an equal effect: the authors observed increase of Treg cells in several organs and decrease of the plaque size [65]. Subcutaneous infusion of apolipoprotein B100 (ApoB100) to ApoE−/− mice was associated with abrogated atherosclerosis development and progression of atherosclerosis, which also appeared to be CD4+CD25+ Treg-dependent [66].

The possibility to achieve stable normalization of the Treg numbers and function through modulation of thymus functional capacity would have been extremely desirable and could help to activate host’s own anti-inflammatory mechanisms to combat chronic low-grade inflammation associated with atherosclerosis.

## 7. Atherosclerosis and Thymus Function: The Shared Effector Molecules

Atherogenesis appeared to be tightly entwined with both immune senescence and inflammaging (over-reactive immune response), each of these conditions having dysfunctional T cells as a hallmark [67]. Since altered function of T lymphocytes’, observed in elderly people, primarily arises from thymic atrophy or thymic involution, the understanding of this process becomes critical for elaboration of the new diagnostic and therapeutic approaches for patients with atherosclerosis. A number of effector molecules and transcription factors appeared to be intricately involved in atherogenesis, metabolic health, and thymic function and could represent the potential new therapeutic targets in atherosclerosis (Table 2).

Interestingly, even though, morphologically, adipocytes in aged thymus resemble adipocytes of the white adipose tissue, the functional properties and molecular pattern in thymic fat are equivalent to those of the beige adipose tissue, considered to be anti-inflammatory and atheroprotective. Molecules typical for beige adipose tissue appear in the thymus in the early stages of ontogenesis and are expressed in TECs. The most important are mitochondrial uncoupling protein (UCP1) and T-box transcription factor (TBX1), the well-known beige-specific markers [72]. UCP1 uncouples oxidative phosphorylation through the leakage of protons across inner mitochondrial membrane and provides metabolically favorable thermogenesis [93]. Presence of UCP1 in the thymus has been revealed both in the whole organ and in the single thymocytes [94,95]. Thymic UCP1 was functional and promoted the production of reactive oxygen species [94]. It most likely played a role in regulation of apoptosis during negative selection of T cells [95]. UCP-1 demonstrated protective properties in atherosclerosis. Perivascular adipose tissue (PVAT), adjacent to big arteries, appeared to have brown phenotype and increased expression of UCP-1 upon injuries [70]. Obesity in mice was associated with reduction of UCP-1 expression along with increase of arterial stiffness and elevation of inflammatory cytokines’ expression in the vessel wall [96]. At the same time, knock-in of *Ucp-1* in pigs alleviated the severity of atherosclerosis in animals. This was most likely achieved through the inhibition of ROS production and activation of NLRP3-inflammasome in PVAT [96].

T-box transcription factor (TBX1) is essential for normal thymus development. *Tbx1* gene haploinsufficiency leads to the development of DiGeorge Syndrome and athymia [71]. Meanwhile, recent findings demonstrated its important role in metabolic regulation [72]. Moreover, TBX1 inhibited mitogen-activated protein kinase-activated protein kinase-2 (MAPKAPK2), the member of intracellular signaling pathways of inflammation, cell differentiation, and apoptosis. LDLR−/− mice with atherosclerosis induced by western diet when supplemented with TBX1 demonstrated improvement of glucose and lipid metabolism, decrease of plaques area, and intraplaque necrosis, as well as stabilization of plaques [73]. Vascular endothelial growth factor (VEGF) interacts with TBX1 and controls its expression [75]. It is yet unknown if the activity of thymic UCP-1 and TBX1 are related to atherogenesis. Investigation of their expression and interconnection with immune function in thymus could have shed light on the impact of thymus beiging on atherogenesis in patients with cardiovascular disorders.

Another ubiquitously expressed molecule, which may be the potential target both in thymus involution and atherosclerosis, is forkhead box protein N1 (Foxn1). Foxn1 is most widely expressed in the thymus and determines differentiation of TECs into cTECs and mTECs [71]. It plays the role in recruitment of hematopoietic cell progenitors to the thymus, differentiation of progenitor cells towards T lymphocytes, and regulation of positive selection [75]. Upregulation of Foxn1 in adult animals led to thymus regeneration [47]. Proatherogenic low density lipoproteins decreased Foxn1 expression in the thymus through low density lipoprotein receptors, enhancing thymus degradation and impairment of immune tolerance [5]. Foxn1 also determines differentiation of adipocytes and regulates diet-induced obesity [76], increasing expression of PPAR-γ, insulin-dependent glucose transporter GLUT4, and IGF2 [77]. In endothelial cells, expression of Foxn1 increased in response to endothelin-1, a marker of endothelial dysfunction, and was associated with oxidative stress [97]. Thus, one should be cautions considering the influence on Foxn1 during atherosclerosis treatment.

Homeobox Protein A3 (HOXA3) represents another transcription factor and is also involved both in thymogenesis and atherosclerosis. Being expressed at the early stages of embryogenesis, it regulates development of thymus, influencing formation of neural crest from mesenchymal stem cells [71,98]. Deletion of HOXA3 led to development of small ectopic thymus and parathyroid glands [99]. At the same time, methylation and the following termination of HOXA3 transcription was typical for early atherosclerotic plaques and was accompanied by infiltration of classical monocytes and inflammatory M1 macrophages into the vessels wall [79]. HOXA3 was responsible for endothelial progenitor cell migration and angiogenesis [80]. Thus, HOXA3 is characterized by pleiotropic effects, which may dampen the development of atherosclerosis both directly and through the modulation of thymus function. Since data of HOXA3 expression in aged thymus are currently absent, as well as potential agents, which may restore its function, further research in this filed is required. Caution should be taken, though, as this transcriptional factor is also involved in the development of various cancers [100].

## 8. Thymosins in Atherosclerosis

Thymosins represent a family of hormone-like peptides, which were first isolated from the calf thymus as early as 1966. Later, it was demonstrated that they are expressed in all cells except red blood cells, but their function varies depending on the type of tissue [101]. Thymosins include three types depending on their isoelectric point, α, β, and γ, with β4 being the most abundant type [47,84].

Thymosin α1 (Tα1) modulates activity of dendritic cells, thus mediating T-dependent immune responses, and promotes immune tolerance through activation of Tregs through IL-10 and indoleamine 2,3-dioxygenase (IDO)1, an immunosuppressive enzyme. Another protolerogenic action of Tα1 is reciprocal regulation of AIRE expression [81]. Ex vivo treatment of blood cells from patients with coronavirus with Tα1 suppressed excessive production of cytokines and inhibited CD8+ cells function [102]. It was demonstrated that Tα1 signals through toll-like receptors (TLR)-9 and TLR-2 [103]. In experimental setting, Tα1 attenuated development of atherosclerotic plaques in rabbits fed a high-cholesterol diet [104]. Preconditioning of the retina with prothymosin α (ProTα) before the induction of ischemia prevented its damage, possibly through the inhibition of the main genes of the most potent inflammatory pathways: myeloid differentiation primary response 88 (MyD88) and Nuclear factor (NF)κB [82]. ProTα was also one of the paracrine factors released by cardiomyocytes after myocardial infarction that recruited endothelial cells to the infarction zone and promoted angiogenesis after ischemia [83].

Another thymosin intricately involved in thymic function regulation is thymosin β4 (Tβ4). It binds cell surface ATP synthase and signals through ATP-responsive purinergic receptor P2X4 [105]. Tβ4 regulates the structure of mTECs and delays age-associated changes in the thymus [84]. An important function of Tβ4, interconnecting it with atherogenesis, is its involvement in angiogenesis. It improves function of endothelial cells, enhancing their proliferative capacity and their ability to form capillary structures and increases capillary density in murine model of hind-limb ischemia [85]. Pretreatment of endothelial progenitor cells with Tβ4 before implantation in the post-infarct myocardium in rats was associated with increased production of VEGF [106]. In acute phase of myocardial infarction, Tβ4 induced anti-inflammatory and antiapoptotic effects [85]. Signaling pathways mediating action of Tβ4 included phosphorylation of Akt, suppression of NF-κB, and activation of intrinsic antioxidant systems [107]. Tβ4 was also involved in regulation of low-density lipoprotein receptor related protein 1 (LRP1) expression on vascular smooth muscle cells (VSMCs) [86].

One might hypothesize that recovery of TEC function through modulation of adipogenesis in thymus could normalize the production of the own host’s thymosins. This could provide desirable beneficial effects of thymosins on the vascular wall without further necessity of their additional external supplementation.

## 9. Retinoic Acid in Thymus and Atherosclerosis

Retinoic acid is a metabolite of vitamin A and can be produced by thymic mesenchymal cells. It turned out to be essential for cTEC and mTEC proliferation and function [87]. Retinoic acid deficiency was associated with decreased thymic cellularity, probably due to higher rates of negative selection mediated by impaired mTECs. In addition, naïve CD8+ T cells failed to undergo activation-induced apoptosis upon TCR-stimulation in the periphery [87]. Retinoic acid performs an inhibitory function in adipogenesis: it suppresses expression of PPAR-γ [108] but at the same time induces expression of UCP-1, thus promoting browning of adipose tissue [109]. Thus, retinoic acid can influence thymus adipogenesis.

In addition, retinoic acid appeared to ameliorate atherosclerosis and induced browning of perivascular adipose tissue, even though the precise mechanisms are not yet identified [88]. It inhibited formation of atheromas in rabbits fed the high-fat diet, prevented restenosis after balloon-angioplasty, and could regulate the function of smooth muscle cells in the vessel wall [89]. Retinoic acid also mediated cholesterol efflux from macrophages, thereby preventing the formation of foam cells [110]. As for neutrophils, which are also involved in pathogenesis of atherosclerosis, retinoic acid induced their polarization towards anti-inflammatory N2-phenotype and inhibited release of neutrophilic extracellular traps (NETs) [90]. Retinoic acid stimulated the function of endogenous NO synthase [91] and decreased expression of endothelin-1 in endothelial cells, alleviating endothelial dysfunction [92]. Currently there are no works in which the protective effects of retinoid acid in atherosclerosis could be attributed to the preservation of thymic function. In our opinion, this represents a promising direction of the research.

## 10. Monitoring of Thymus Function

The routine analysis of immune function is usually focused on peripheral blood. However, the function of thymus may also be evaluated directly [4].

T cells undergo recombination of T cell receptor (TCR) upon maturation in the thymus. It occurs through the unique process specific only for lymphoid cells: excision of fragments of genes responsible for TCR synthesis, thus creating a large number of TCR variants, which are able to recognize the vast majority of antigens [111]. Fragments of excised DNA remain in naïve T cells outside the nucleus and may be well tracked using real-time polymerase chain reaction (PCR) [112]. The number of signal joint T cell receptor excision circles (sjTRECs) per mg of thymus tissue or per definite number of T lymphocytes (either 10^5^ or 10^6^ cells) reflects the thymopoetic capacity of the thymus [111]. It has been recently demonstrated that patients with unstable angina were characterized by the decreased number of sjTREC copies compared to stable patients and age-matched controls, and lower sjTREC values were typical for patients with more severe atherosclerosis [43]. Low sjTREC levels were also associated with increased risk of mortality in elderly patients [113].

Another approach to evaluation of thymic function is determination of the ratio between naïve CD45RA+ lymphocytes (CD45RA is the marker molecule of all the naïve leukocytes, which become cleaved during activation and turns into CD45RO) vs. activated CD62L+ lymphocytes and expression of CD31 molecule [4]. CD31+CD45RA+CD45RO– cells appeared to be enriched with TREC-containing cells and can be used for diagnostics both among the elderly patients and children with lymphopenic conditions [114]. Decreased expression of CD31 on Treg-lymphocytes from patients with coronary artery disease was associated with compromised Treg function [43].

At the macro-level, computed tomography (CT) performed at the area of the anterior mediastinum allows us to reveal the decrease of the CT attenuation value in the aging thymus due to the replacement of the epithelial tissue with adipose tissue. A special scoring system was elaborated where the degree of fat tissue replacement was evaluated, with complete fat tissue substitution being equal to Score 0 and solid thymus without adipose tissue being equal to Score 3. According to this system, among patients of median age 58.9 years, higher scores were typical to women and lower scores were associated with cigarette smoking and higher BMI [115]. After the age of 50, CT attenuation value reached the plateau and CT was unable to detect the remaining islets of the functioning thymic tissue [116]. Other imaging modalities available for study of thymus function include magnetic resonance imaging (MRI), X-ray, and ultrasound. MRI appeared to be essential for differentiation between normal and hyperplastic thymus, allowing us to establish the presence of macroscopic and intracellular fat [117], while ultrasound represents a method of choice in pediatric patients [118].

## 11. Future Perspectives and Conclusions

The evaluation of thymus functional state has a promising diagnostic and therapeutic potential in patients with atherosclerosis. This focus is highly understated at the moment.

Keeping in mind that the thymus shares transcription factors [72] and signaling pathways [119] with metabolically active tissues, including brown adipose tissue, therapeutic approaches directed to these factors could improve the state of both immune regulation and metabolic impairments. Such pleiotropic effects could reduce both the residual inflammatory and residual metabolic risks in patients with disorders associated with atherosclerosis [120] (Figure 3). However, caution should be taken in studies targeting signaling pathways common to thymus involution and atherosclerosis as all the agents reducing atherosclerosis and increasing immune tolerance have side effects associated with increased predisposition to cancer.

Nearly half of the human genome is represented by transposable elements (TE), the repetitive sequences that are either spread across the genome (the long and short interspersed nuclear elements (LINE and SINE)) or terminate the transcription of the genes (the long terminal repeats; LTR). mTECS appeared to express high levels of TE, which could be translated and are expressed as MHC-associated proteins. There is a hypothesis that these TE may play the role in sustenance of immune tolerance [121]. TEs are currently regarded as therapeutic targets in anti-cancer therapy [122]. A possibility exists that some therapeutic approaches may be elaborated targeting TE to improve immune tolerance in atherosclerosis. Perhaps we need to specifically target mTECs for this purpose (Figure 3).

Surprisingly, thymus adipose tissue (TAT) in the elderly patients (individuals older than 70 years) was characterized by increased angiogenic properties. In particular, it produced more VEGF than TAT of the middle-aged patients and subcutaneous adipose tissue (SAT), which was dependent upon signaling of microRNAs (miRNAs) miR-15b-5p and miR-29a-3p. This property of TAT may be used to modify neovascularization in patients with major adverse cardiovascular events [123]. Heart surgery, accompanied by sternotomy, is inevitably accompanied by TAT excision and removal. Hybrid surgery followed by isolation of TAT and future culture might serve to improve outcome in patients with coronary artery disease (Figure 3).

On the other hand, thymic function appeared to be critical during heart transplantation. In this situation, more preserved and functional thymus was associated with higher frequency of heart rejection due to the higher frequencies of recent thymic CD4+ migrants in myocardium [124]. Thus, thymectomy should be considered for the prevention of alloimmune reactions in patients undergoing heart transplantation.

Meanwhile, progress has been made in thymic tissue transplantation, based on the preliminary dilution of the donor T cells and transplantation of the thymic stroma. Even though a certain number of complications were observed, including development of autoimmune disorders, graft-versus-host-disease, and infections, the survival rate was equal to 77%, which is very inspiring [125]. There is a promising approach of co-transplantation of thymus and another donor organ, including heart, to reduce the risk of organ rejection. In this case, T cells from the donor thymus tissue are being depleted during in vitro culture, but the niche, consisting of TECs and stromal cells, required for the growth of the recipient T cells, remain intact and may favor the restoration of the recipient’s immune system [126,127]. The question remains, if such revolutionary approaches can benefit patients with advanced atherosclerosis.

Thus, addressing the role of thymus during atherosclerosis expands the knowledge of atherogenesis far beyond cholesterol-based theory, which, even though has undoubtedly been a great breakthrough in medicine more than a century ago, in certain cases raises more questions than provides answers [128]. Including thymus in the contemporary scenario of atherosclerosis not as a “silent player”, but rather as one of the “leading characters” could shed a new light on its pathogenesis and provide additional tools of its attenuation and prevention of complications.

## Figures and Tables

**Figure 1 biomedicines-12-01408-f001:**
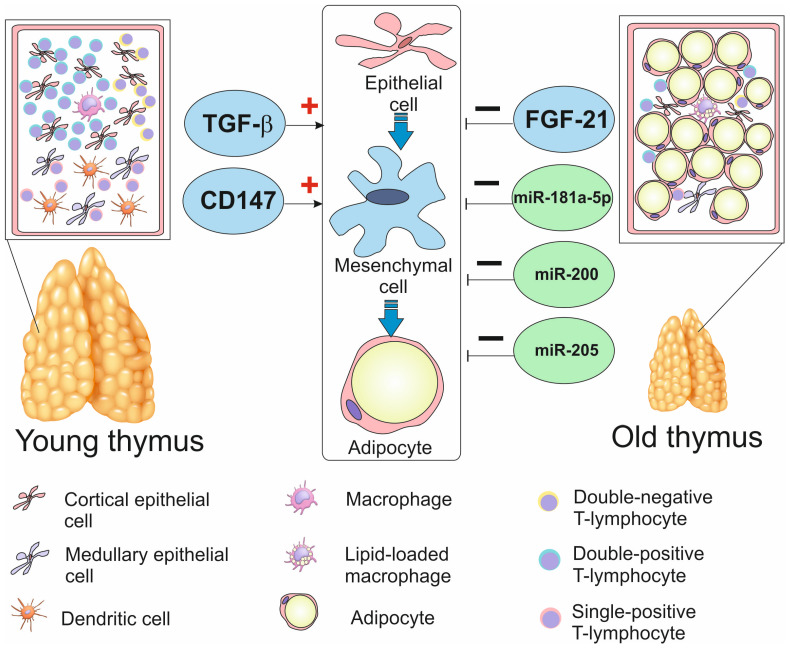
Thymus adipose involution through epithelial-mesenchymal transition. The young thymus consists of well-defined lobules. Each lobule consists of subcapsular, cortical, and medullary zones, while the old thymus is majorly replaced by adipose tissue, containing only the small remaining islets of thymopoiesis. The replacement of thymus with adipose tissue most likely occurs through epithelial-mesenchymal transition: thymic epithelial cells differentiate into mesenchymal cells, which ultimately differentiate into adipocytes. Lipid-loaded macrophage appear in thymus Transforming growth factor (TGF)-β and molecule CD147, expressed on T lymphocytes, favor epithelial-mesenchymal transition in thymus, while fibroblast growth factor (FGF)-21 and microRNAs miR-181a-5p, miR-200, and miR-205 suppress it. The image of thymus used here and in other figures was purchased from VectorStock.

**Figure 2 biomedicines-12-01408-f002:**
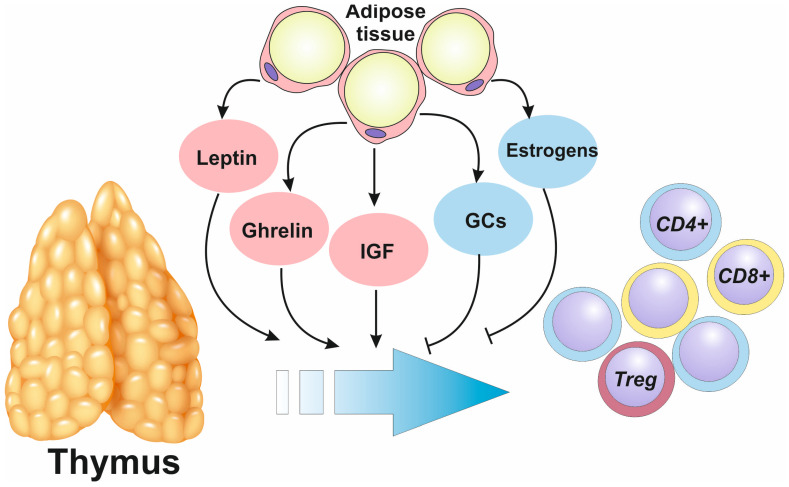
Regulation of thymic function by adipokines. Functional activity of thymus, consisting of generation and maturation of the competent T lymphocytes, depends on molecules produced by adipose tissue (adipokines). Leptin, ghrelin and insulin growth factor (IGF) potentiate thymopiesis, while glucocorticoids (GCs) and estrogens inhibit thymopiesis. Treg—T regulatory lymphocytes; CD8+—naïve cytotoxic CD8+ T lymphocytes; CD4+—naïve helper CD4+ T lymphocytes.

**Figure 3 biomedicines-12-01408-f003:**
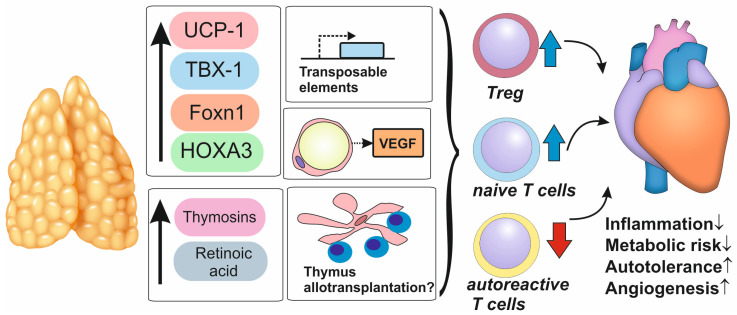
Potential therapeutic approaches associated with thymus in atherosclerosis. The possible approaches to modulate thymic function in patients with atherosclerosis include (1) upregulation of the transcription factors, involved in regulation of the thymus functional activity (uncoupling protein (UCP1); T-box transcription factor (TBX1); forkhead box protein N1 (Foxn1); homebox protein A3 (HOXA3)); (2) supplementation or activation of the production of thymus effector molecules (thymosins; retinoic acid); (3) targeting thymus transposable elements; (4) use of thymus adipose tissue for production of angiogenic molecules; (5) use of allogenic thymus transplants to improve maturation of the hosts own T lymphocytes. The expected result is increase of the thymic output of the natural T regulatory lymphocytes (Tregs), increase of the functional naïve T lymphocytes and decrease of autoreactive T lymphocytes’ numbers in the periphery, accompanied by the reduced inflammatory and metabolic risks, potentiation of self-tolerance and angiogenesis.

**Table 1 biomedicines-12-01408-t001:** Protective functions of thymus in atherosclerosis.

Protective Mechanism	Result of Failure	Object of Study	References
Negative selection of auto-specific T lymphocytes	1. Diminished expression of tissue restricted antigens in thymus (ApoB-antigen) 2. Decreased frequency of cells undergoing clonal deletion in thymus 3. Appearance of ApoB-specific T lymphocytes in circulation	C57BL/6J mice (males and females; young and aged animals)	Hester A.K. et al. (2022) [11]
Maintenance of T cell homeostasis	1. Decreased numbers of recent thymic migrants in patients with non-ST elevation acute coronary syndrome (NSTACS) compared to stable angina and control (according to the signal joint T cell receptor excision circles number) 2. Decreased numbers of recent thymic migrants in severer atherosclerosis compared to less severe atherosclerosis (according to the signal joint T cell receptor excision circles number)	CAD patients	Huang S. et al. (2016) [43]
	1. Decreased frequency of recent thymic migrants in patients with acute coronary syndrome compared to healthy controls (according to the numbers of CD31+ cells) 2. Decreased production of anti-inflammatory TGF-β and IL-10	CAD patients	Huang L. et al. (2017) [49]
Generation of atheroprotective natural T regulatory cells	Development of larger plaques	ApoE−/− mice depleted of Tregs	Ait-Oufella H. et al. (2006) [50]
	1. 2.1-fold increase in the size of atherosclerotic lesions 2. 1.7-fold increase in plasma cholesterol 3. Increased levels of very low density lipoproteins cholesterol 4. Altered systemic lipid metabolism	LDLR−/− mice depeleted of Tregs	Klingenberg R. et al. (2013) [51]
	1. Decreased CD4(+)CD25(+)CD127(low)CD45RO(−)CD45RA(+)CD31(+) numbers of recent thymic emigrant Treg cells in NSTACS patients compared to patients with chronic stable angina and chest pain syndrome 2. Decreased numbers of Treg cells with T cell receptor excision circle in NSTACS patients 3. Increased apoptosis in Treg cells 4. Increased markers of inflammation (TNF concentration; decreased IL-10/TNF ratio	CAD patients	Zhang W.C. et al. (2012) [52]
	1. Low levels of CD4+FoxP3+ cells associated with development of myocardial infarction 2. Increased release of IL-2, IL-6, IL-8, IFN-γ, TNF-α and IL-1β in patients with low numbers of Treg cells	Participants of Malmö Diet and Cancer Study	Wigren M. et al. (2012) [53]
	1. Decreased numbers of thymic Helios+ Treg cells in patients with acute coronary syndrome compared to patients with stable angina and control. 2. Negative correlation between CD4+Foxp3+Helios+ Tregs and IL-6 and positive correlation with circulating TGF-β and HDL-C in CAD patients and control	CAD patients	Jiang L. et al. (2017) [54]
Secretory activity of thymic adipocytes	Increased secretion of leptin and decreased secretion of insulin are associated with increased arterial stiffness	CAD patients undergoing CABG	Naryzhnaya N.V. et al. (2020) [48]

ApoB—apolipoprotein B; CABG—coronary artery bypass graft; CAD—coronary artery disease; CD—cluster of differentiation; HDL-C—high-density lipoproteins cholesterol; IFN—interferon; IL—interleukin; NSTACS—non-ST elevation acute coronary syndrome; TGF—transforming growth factor; TNF—tumor necrosis factor; Treg—T regulatory lymphocytes.

**Table 2 biomedicines-12-01408-t002:** Shared transcription factors and effector molecules in thymus and atherosclerosis.

Title	Function	Role in Thymus	Role in Atherogenesis	Activation/Inhibitory Agents	References
Mitochondrial uncoupling protein (UCP1)	Uncouples oxidative phosphorylation through the leakage of protons across inner mitochondrial membrane and provides thermogenesis	Regulation of positive and negative selection through apoptosis	Browning of perivascular adipose tissue, switch to anti-inflammatory phenotype of adipose tissue	β3-adrenoreceptor agonists; catecholamines, fibroblast growth factor (FGF)−21, thyroid hormones activate	Kang G.S. et al. (2023) [68]; Adams A.E. et al. (2010) [69]; Adachi Y. et al. (2022) [70]
T-box transcription factor (TBX1)	Transcription factor, controlling development of organs and tissue; target genes are not known	Critical for development of normal thymus (athymia and DiGeorge Syndrome in case of absence)	Browning of adipose tissue; inhibition of intracellular signaling pathways of inflammation, cell differentiation and apoptosis	Vascular endothelial growth factor activates	Giardino G. et al. (2020) [71]; Banfai K. et al. (2019) [72]; Ozcan L. et al. (2021) [73]; Stalmans I. et al. (2003) [74]
Forkhead box protein N1 (Foxn1)	Transcription factor. Controls expression of threonine peptidases, components of the proteasome complex, protein transporters and CD83	1. Recruitment of hematopoietic cell progenitors to thymus and their differentiation towards T lymphocytes 2. Regulation of positive selection	1. Increases pro-adipogenic factors PPAR-γ, insulin-dependent glucose transporter GLUT4 and IGF2 and favors obesity 2. Induces oxidative stress in endothelial cells	LDL-C inhibits; Endothelin-1 activates	Žuklys S. et al. (2016) [75]; Dai X. et al. (2018) [5]; Walendzik K. et al. (2020) [76]; Gawronska-Kozak B. et al. (2021) [77]
Homeobox Protein A3 (HOXA3)	Transcription factor. Regulates morphogenesis and cellular differentiation; predictive targets include 273 genes	Regulates early stages of development in thymus	1. Controls development of M1/M2 macrophages in vessel wall; 2. Controls endothelial cell migration and angiogenesis	Retinoic acid	Giardino G. (2020) [71]; Diman N.Y. et al. (2011) [78]; Xu C. et al. (2022) [79]; Mace K.A. et al. (2005) [80]
Thymosin α1 (Tα1)	28 amino-acid peptide, immune modulator	1. Regulation of AIRE expression; 2. Activation of tolerogenic dendritic cells	1. Inhibition of pro-inflammatory signaling pathways in ischemia; 2. Recruitment of endothelial cells; angiogenesis 3. Recombinant substance exists	?	Moretti S. et al. (2015) [81]; Halder S.K. et al. (2015) [82]; Gladka M.M. et al. (2021) [83]
Thymosin β4 (Tβ4)	43 amino-acid peptide, G-actin sequestering peptide	1. Regulates differentiation of thymocytes through cytoskeletal rearrangement of thymus epithelial cells	1. Improves function of endothelial cells; 2. Anti-inflammatory action in myocardial infarction 3. Regulates expression of low density lipoprotein receptor related protein 1 expression on vascular smooth muscle cells 4. Recombinant substance exists	?	Ying Y. et al. (2024) [84]; Xing Y. et al. (2021) [85]; Munshaw S. et al. (2023) [86]
Retinoic acid	Metabolite of vitamin A	1. Limits the rate of negative selection in thymus	1. Induces browning of perivascular adipose tissue 2. Prevents formation of foam cells 3. Reduces the size of atherosclerotic lesions 4. Inhibits activation of inflammatory neutrophils 5. Stimulates production of endogenous NO 6. Decreases expression of endothelin-1 by endothelial cells	1. Dietary vitamin A 2. Short-chain retinol dehydrogenases (catalyze synthesis of retinoic acid from retinol)	Wendland K. et al. (2018) [87]; Kalisz M. et al. (2021) [88]; Deng Q., Chen J. (2022) [89]; Cai W. et al. (2019) [90]; Achan V. et al. (2002) [91]; Yokota J. et al. (2001) [92]

CD—cluster of differentiation; GLUT4—glucose transporter type 4; IGF—insulin growth factor; NO—nitric oxide; PPAR-γ—Peroxisome proliferator-activated receptor; question mark in the table (?) indicates that no data were discovered.

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
