# Peer review of "Thymus in Cardiometabolic Impairments and Atherosclerosis: Not a Silent Player?"

_biomedicines, 2024, doi:10.3390/biomedicines12071408_

Round 1

Reviewer 1 Report

Comments and Suggestions for Authors

The review article Thymus in cardiometabolic impairments and atherosclerosis: not a silent player highlights an intriguing and sometimes controversial topic within the CVD field, and so this topic is welcome.  Overall, the manuscript appears to be well-written, formatted/sectioned adequately, and flows appropriately.  Major reviewer comments, that the authors should heed to, are listed below:

*Please change keywords to words, statements, and phrases not already included within the title and abstract.

*Include reputable citations and accompanied text on thyroidectomy patients having altered CVD risk.

*Inflammation appears to be a double-edged sword regarding CVD and so discussing how some types of inflammation may appear to be cardioprotective should also be discussed and highlighted.

*Please spell-out all abbreviations when first used for terms that may be unfamilar to your audience.

*Figure 1 is vague and ambigious without an adequate legend to guide the reader through this figure, so please include this text.

*A major limitation of this review is inadequate figure and table number.  Suggestions to enrich figures/tables would be to detail atheroprotective mechanisms of the thymus related to atherosclerosis (moreso than in lone Fig. 1), contrast young thymus from old thymus, list all known thymosins and known functions, and describe how the thymus modulates inflammation.

*Please combine Future Perspectives and Conclusions into one section.

Comments on the Quality of English Language

English is okay for an initial submission.

Author Response

Dear reviewer!

We sincerely thank you for your time and efforts that you spent working on our manuscript. We are grateful for all the detailed suggestions you have made, and tried our best to address them. Please, find the elaborated answers to your comments below. We have marked the introduced changes with blue color, where it was possible.

*Please change keywords to words, statements, and phrases not already included within the title and abstract.

We have replaced the key words already mentioned in the title and abstract with new settings of key words. The replacing key words are: thymus atrophy; thymus adipose tissue; atherosclerotic plaque; atherogenesis. 

*Include reputable citations and accompanied text on thyroidectomy patients having altered CVD risk.

As we have understood correctly, this comment concerned thymectomy in patients. Unfortunately, we have found no works aimed to evaluate the CVD risk in patients with thymectomy. We emphasized it in the end of the chapter “Thymus and atherosclerosis”, as a fact that underscores the necessity for such studies to be performed. The purpose of the current review is to attract attention of the scientists including clinical scientists to the study of the role of thymus in atherosclerosis.

 *Inflammation appears to be a double-edged sword regarding CVD and so discussing how some types of inflammation may appear to be cardioprotective should also be discussed and highlighted.

Thank you for this valuable comment. We decided to add the chapter “Thymic T regulatory lymphocytes control atherogenesis”, which addresses the involvement of anti-inflammatory T regulatory lymphocytes, which have primarily thymic origin, in the control of atherogenesis. Clinical trials devoted to Treg lymphocytes in patients with CVD, including acute events, were also reviewed in this section.

*Please spell-out all abbreviations when first used for terms that may be unfamilar to your audience.

We apologize that not all the abbreviations were spelled-out in the original submission. We tried to our best to address this drawback.

 *Figure 1 is vague and ambigious without an adequate legend to guide the reader through this figure, so please include this text.

We decided to leave Figure 1 as a graphical abstract. 

*A major limitation of this review is inadequate figure and table number.  Suggestions to enrich figures/tables would be to detail atheroprotective mechanisms of the thymus related to atherosclerosis (moreso than in lone Fig. 1), contrast young thymus from old thymus, list all known thymosins and known functions, and describe how the thymus modulates inflammation.

Thank you for your valuable suggestion!

We have added 2 tables: Table 1. Protective functions of thymus in atherosclerosis and Table 2. Shared transcription factors and effector molecules in thymus and atherosclerosis. And 3 Figures: Figure 1. Thymus adipose involution through epithelial-mesenchymal transition; Figure 2. Regulation of thymic function by adipokines and Figure 3. Potential of therapeutic approaches associated with thymus in atherosclerosis.

*Please combine Future Perspectives and Conclusions into one section.

Thank you! We have combined these sections.

Reviewer 2 Report

Comments and Suggestions for Authors

Quite massive preclinical evidence supports the immune-mediated contribution to the atherogenic process although it should be fair to recognize that the results obtained insofar from clinical intervention trials is limited and controversial (see ref 9 and 10 cited in this document) and awaits more solid ground. In this regard, even sparser clinical evidence supports the role of  T lymphocytes in the clinical phenotypes of cardiovascular disease of atherosclerotic nature and even less as regards the specific role of thymus. In this context the admittedly quite compelling recent evidence (see ref 22 in this document) on the dire consequences of thymectomy appears to involve mainly cancer-related deaths. The Authors should also recognize that the causal link between the aging thymus and both cardiovascular and metabolic diseases in humans, at least that emerging by the data presented herein, is at best circumstantial although certainly worth of further evaluation mainly in the clinical field.

In conclusion, I find this paper too long, inconclusive and very difficult to be read even by somewhat literate readers in the cardiovascular field to whom, I assume, a review like this should be primarily aimed at. Certainly does not help the reference to a congeries of biological factors cited as acronyms (e.g. Nlrp3, FGF21, Fezf2, Wnt4, LAP2α, PLIN+PPARγ2+, PLIN+UCP1+, PLIN+Iba-1+, PLIN+NG2+ and so on) throughout the text without any presentation of their function. For this reason, I tend to recommend the rejection of this paper but I leave to the Editor(s) this decision.

Author Response

Dear reviewer!

We sincerely thank you for your time and efforts that you spent working on our manuscript. We are grateful for the comments you have made, and tried our best to address them. We have marked the introduced changes with blue color, where it was possible.

We have shortened the article: (1) excluded the chapter “Atherosclerosis as a T cell-mediated disease”; (2) combined information devoted to transcription factors into 1 chapter, leaving only the important facts concerning these factors; (3) shortened over-complicated parts of the manuscript.

We have also added tables and figures, which, we are hoping, will make the review easier understandable: Table 1. Protective functions of thymus in atherosclerosis; Table 2. Shared transcription factors and effector molecules in thymus and atherosclerosis; Figure 1. Thymus adipose involution through epithelial-mesenchymal transition; Figure 2. Regulation of thymic function by adipokines and Figure 3. Potential of therapeutic approaches associated with thymus in atherosclerosis.

We have also changed the title, to stress that the subject of the review requires further studies.

 The Review:

Quite massive preclinical evidence supports the immune-mediated contribution to the atherogenic process although it should be fair to recognize that the results obtained insofar from clinical intervention trials is limited and controversial (see ref 9 and 10 cited in this document) and awaits more solid ground. In this regard, even sparser clinical evidence supports the role of  T lymphocytes in the clinical phenotypes of cardiovascular disease of atherosclerotic nature and even less as regards the specific role of thymus. In this context the admittedly quite compelling recent evidence (see ref 22 in this document) on the dire consequences of thymectomy appears to involve mainly cancer-related deaths. The Authors should also recognize that the causal link between the aging thymus and both cardiovascular and metabolic diseases in humans, at least that emerging by the data presented herein, is at best circumstantial although certainly worth of further evaluation mainly in the clinical field.

In conclusion, I find this paper too long, inconclusive and very difficult to be read even by somewhat literate readers in the cardiovascular field to whom, I assume, a review like this should be primarily aimed at. Certainly does not help the reference to a congeries of biological factors cited as acronyms (e.g. Nlrp3, FGF21, Fezf2, Wnt4, LAP2α, PLIN+PPARγ2+, PLIN+UCP1+, PLIN+Iba-1+, PLIN+NG2+ and so on) throughout the text without any presentation of their function. For this reason, I tend to recommend the rejection of this paper but I leave to the Editor(s) this decision.

Reviewer 3 Report

Comments and Suggestions for Authors

Manuscript is relatively well written, but too long. My suggestion is to reconsider the length of paragraph 4 (Atherosclerosis as a T cell-mediated disease), since appears not focused on thymus. Also, please avoid the use of acronyms in subtitles. Figure 1 should be used as graphical abstract, rather than illustration. 

Comments on the Quality of English Language

Minor editing of English language required

Author Response

Dear reviewer!

We sincerely thank you for your time and efforts that you spent working on our manuscript. We are grateful for the comments you have made, and tried our best to address them. We have marked the introduced changes with blue color, where it was possible.

We have shortened the article: (1) excluded the chapter “Atherosclerosis as a T cell-mediated disease”; (2) combined information devoted to transcription factors into 1 chapter, leaving only the important facts concerning these factors; (3) shortened over-complicated parts of the manuscript.

We have also added tables and figures, which, we are hoping, will make the review easier understandable: Table 1. Protective functions of thymus in atherosclerosis; Table 2. Shared transcription factors and effector molecules in thymus and atherosclerosis; Figure 1. Thymus adipose involution through epithelial-mesenchymal transition; Figure 2. Regulation of thymic function by adipokines and Figure 3. Potential of therapeutic approaches associated with thymus in atherosclerosis.

The Figure 1 in the original version of the manuscript was used as a graphical abstract, according to your suggestion.

The Review:

Manuscript is relatively well written, but too long. My suggestion is to reconsider the length of paragraph 4 (Atherosclerosis as a T cell-mediated disease), since appears not focused on thymus. Also, please avoid the use of acronyms in subtitles. Figure 1 should be used as graphical abstract, rather than illustration. 

Round 2

Reviewer 2 Report

Comments and Suggestions for Authors

I am glad to recognize the significant improvement of the revised document